



# On the circulation, water mass distribution, and nutrient concentrations of the western Chukchi Sea

Jaclyn Clement Kinney[1], Karen M. Assmann[2,3], Wieslaw Maslowski[1], Göran Björk[2], Martin Jakobsson[4], Sara Jutterström[5], Younjoo J. Lee[1], Robert Osinski[6], Igor Semiletov[7,8,9], Adam Ulfsbo[2], Irene Wåhlström[10], Leif G. Anderson[2]

[1]Naval Postgraduate School, 833 Dyer Rd., Monterey, CA, 93943, USA
[2]Department of Marine Sciences, University of Gothenburg, 405 30 Gothenburg, Sweden
[3]Institute of Marine Research, P.O. Box 6606 Langnes, NO-9296 Tromsø, Norway
[4]Department of Geological Sciences, Stockholm University, 106 91 Stockholm, Sweden
[5]IVL Swedish Environmental Research Institute, Box 530 21, 400 14 Gothenburg, Sweden
[6] Institute of Oceanology, Polish Academy of Sciences, 81-712 Sopot, Poland
[7]International Arctic Research Center, University Alaska Fairbanks, Fairbanks, AK 99775, USA
[8]Pacific Oceanological Institute, Russian Academy of Sciences Far Eastern Branch, Vladivostok 690041, Russia
[9]The National Research Tomsk Polytechnic University, Tomsk, Russia
[10]Swedish Meteorological and Hydrological Institute, Norrköping, Sweden

*Correspondence to*: Jaclyn Clement Kinney (jlclemen@nps.edu) and Leif G. Anderson (leif.anderson@marine.gu.se)

**Abstract.** Substantial amounts of nutrients and carbon enter the Arctic Ocean from the Pacific Ocean through Bering Strait, distributed over three main pathways. Water with low salinities and nutrient concentrations takes an eastern route along the Alaskan coast, as Alaskan Coastal Water. A central pathway exhibits intermediate salinity and nutrient concentrations, while the most nutrient-rich water enters Bering Strait on its western side. Towards the Arctic Ocean the flow of these water masses is subject to strong topographic steering within the Chukchi Sea with volume transports modulated by the wind field. In this contribution we use data from several sections crossing Herald Canyon collected in 2008 and 2014 together with numerical modeling to investigate the circulation and transport in the western part of the Chukchi Sea. We find that a substantial fraction of water from the Chukchi Sea enters the East Siberian Sea south of Wrangel Island and circulates in an anticyclonic direction around the island. This water then contributes to the high nutrient waters of Herald Canyon. The bottom of the canyon has the highest nutrient concentrations, likely as a result of addition from the degradation of organic matter at the sediment surface in the East Siberian Sea. The flux of nutrients (nitrate, phosphate, and silicate) and dissolved inorganic carbon in Bering Summer Water and Winter Water is computed by combining hydrographic and nutrient observations with geostrophic transports referenced to LADCP and surface drift data. Even if there are some general similarities between the years, there are differences in both the temperature-salinity and nutrient characteristics. To assess these differences, and also to get a wider temporal and spatial view, numerical modeling results are applied. According to model results, high frequency variability dominates the flow in Herald Canyon. This leads us to conclude that this region needs to be monitored over a longer time frame to deduce the temporal variability and potential trends.



# 1 Introduction

The Arctic Ocean has experienced large changes in recent decades with a decrease in summer sea ice cover as the most prominent, impacting a number of processes including the functioning of the ecosystem (e.g. Meier et al., 2014; Kwok, 2018). More open water decreases the albedo which amplifies warming (Kashiwase et al., 2017). To some degree this change in albedo is compensated for by more cloud formation caused by increased evaporation from the open water. The melting of sea

ice is mainly caused by atmospheric forcing (Ding et al., 2017), but inflow of warm surface water from the Atlantic and Pacific Oceans (see Fig. 1a for the general circulation), also plays a role, as well as the heat mixed up into the surface ocean from deeper layers (e.g. Polyakov et al., 2013, 2017; Stroeve and Notz, 2018). The temperature of the Atlantic Water entering the Arctic Ocean through Fram Strait has varied over the last decade with an increasing trend overall (e.g. Wang et al., 2020). On the other hand, the three-dimensional structure of the flow and property transport through Bering Strait is much less known

(Clement Kinney et al., 2014). One reason for this deficit is the difficulty of having sustained measurements in Russian waters. Limited long-term mooring records between 1990-2019 indicate an increase in volume transport as well as a winter freshening and spring warming (Woodgate, 2018; Woodgate and Peralta-Ferriz, 2021).

One essential question related to Arctic warming is how the regional system might feedback to the global climate system.

These feedbacks could either be changes in ocean circulation, notably deep water formation and its impact on the thermohaline circulation, the changes in albedo with decreasing summer sea ice coverage (Wang et al., 2020), or through changes in the sources and sinks of greenhouse gases like carbon dioxide and methane. The marine sink of carbon dioxide is determined by ocean circulation, but also by changes in the ecosystem, including primary productivity that has recently been reported to intensify due to elevated light conditions when the sea ice vanishes or gets thinner (e.g. Arrigo and van Dijken, 2015; Clement

Kinney et al., 2020). Arctic Ocean primary production is often limited by nutrient supply (e.g. Anderson et al., 2003; Mills et al., 2018) with external sources from the Atlantic and Pacific oceans, as well as river runoff. Furthermore, nutrients can be supplied to the surface water from below by mixing that, in turn, can promote primary productivity (Williams and Carmack, 2015). However, the nutrients in the subsurface waters often have a source of organic matter mineralization, a process that also produces carbon dioxide. Thus, this path of primary production might have less impact on the oceanic carbon sink,

stressing the need to assess the contribution of the external sources. The Pacific inflow has higher nutrient concentrations than that from the Atlantic (e.g. Wilson and Wallace, 1990), but has a significant deficit in nitrate relative to phosphate due to denitrification in the upstream regions (e.g. Kaltin and Anderson, 2005). Additional denitrification also occurs in the Arctic shelf seas, thus further contributing to this nitrate deficit (Anderson et al., 2011).

Pacific Water enters the Chukchi Sea through Bering Strait (Fig. 1a) in three water masses of different properties: Alaskan Coastal Water, Bering Shelf Water, and Anadyr Water (e.g. Coachman et al., 1975). Of these, Bering Shelf Water and Anadyr Water partly mix north of Bering Strait to form what is known as Bering Sea Water (e.g. Pisareva et al., 2015). These water





masses largely take different paths in the Chukchi Sea before entering the deep central Arctic Ocean (Brugler et al., 2014)
(Fig. 1b), but mixing between these and other waters, like the water of the Siberian Coastal Current (SCC) occur (e.g.
Weingartner et al., 1999). In general, Alaskan Coastal Water follows the coast towards Barrow Canyon, while Bering Summer
Water (also known as Bering Sea Water) splits into two branches, with one that flows through the central channel over Hanna
Shoal, and one that takes a more westerly path and leaves through Herald Canyon. Part of the latter also flows into the East
Siberian Sea through Long Strait. These three water masses have different salinities and nutrient concentrations (Walsh et al.,
1989), where Anadyr Water is the saltiest and also has the highest nutrient concentration. Anadyr Water tends to be found on
the western side of Bering Strait and flows through Long Strait.

Water flowing from the Pacific Ocean into the Arctic Ocean affects the hydrography as well as the ecosystems, both in the
neighboring shelf seas and the Beaufort Gyre of the Canada Basin. In this contribution, the pathways of Pacific Ocean water
in the western Chukchi Sea, as well as its interaction with the waters of the eastern East Siberian Sea, are investigated based
on both observations and numerical modelling. The resulting volume transport is used to assess the supply of nutrients to the
Arctic Ocean through Herald Canyon.

## 2 Methods

### 2.1 Data

The data presented were collected during two cruises, the International Siberian Shelf Study in 2008 (ISSS-08) and the Swedish
– Russian – US Arctic Ocean Investigation of Climate-Cryosphere-Carbon Interactions (SWERUS-C3) expedition in 2014,
see Fig. 1b for station locations. The ISSS-08 cruise was conducted along the Siberian shelf seas using the Russian vessel
*Yacob Smirnitskyi* and started on 15 August 2008 in Kirkenes, Norway, and ended in the same port on 26 September. The
SWERUS-C3 expedition was conducted along the continental shelf break of northern Siberia. The expedition consisted of two
legs with icebreaker *Oden*. Leg 1 started on 5 July 2014 in Tromsø, Norway, and followed the Siberian continental shelf to
end in Barrow, Alaska, 21 August. Leg 2 focused on the continental shelf break, slope and the adjacent deep Arctic Ocean
basin and ended in Tromsø on 3 October. The Herald Canyon stations were occupied during 6-8 September in 2008 and during
24-26 August in 2014. To assess the atmospheric and sea ice conditions during and preceding the two cruises we used daily
NSIDC SMMR sea ice concentrations (Cavalieri et al., 1996) at 25 km resolution and daily 10 m wind velocities and monthly
surface heat fluxes from the NCEP North American Regional Analysis (NARR; Mesinger et al., 2006) at 32 km resolution.

During both cruises CTD observations were made using a Seabird 911+ CTD equipped with dual Seabird SBE3 temperature,
SBE4C conductivity and SBE43 oxygen sensors. Water samples were collected from a CTD/rosette system equipped with 12
and 24 bottles of Niskin type, respectively, each having a volume of 7 L. The salinity data was calibrated against water samples





analyzed onboard using a laboratory salinometer (Autosal, Guildline Instruments). The calibration and processing procedures

for the 2014 SWERUS-C3 data are described in Björk et al. (2018).

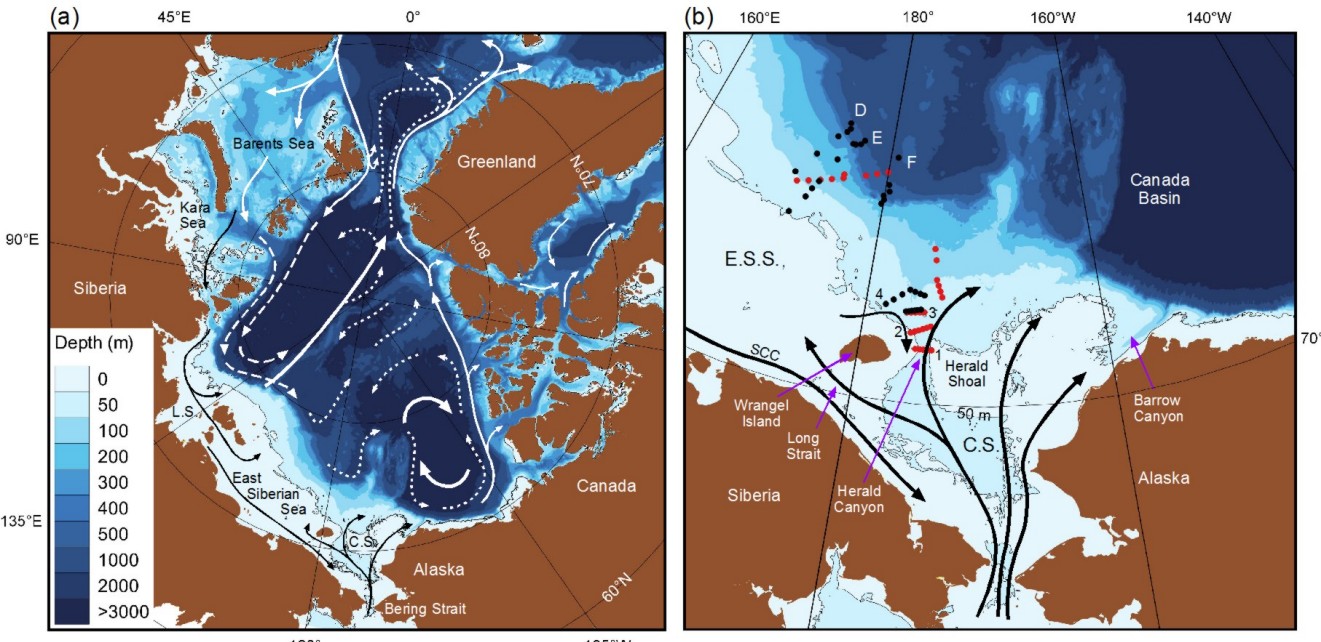

**Figure 1: Map of the Arctic Ocean (a) with the mean schematic oceanic circulation illustrated by solid arrow for surface currents, interrupted arrows for intermediate currents, and dotted arrows for deep currents. In (b) a close up of the eastern East Siberian Sea (E.S.S.) and Chukchi Seas (C.S.), with stations noted from the ISSS-08 (red dots) and SWERUS-C3 (black dots). Herald Canyon**

**sections are labelled with 1 to 4 and the continental slope sections labelled D, E, and F. SCC denotes the Siberian Coastal Current. Bathymetry is from the International Bathymetric Chart of the Arctic Ocean (IBCAO) Version 4.0 (Jakobsson et al. 2020). Please note black dots overlying red ones in section 3.**

## 2.1.1. Current velocities and transports

Two 300 MHz RDI Workhorse Acoustic Doppler Current Profilers (ADCP) were mounted on the CTD rosette during the

SWERUS-C3 cruise as an upward and downward looking pair. The Lowered ADCP (LADCP) data was processed with the

Lamont-Doherty Earth Observatory software package (Thurnherr et al., 2010). The velocities were de-tided (removing the

tidal component) using the Arctic Ocean tidal model AOTIM (Padman and Erofeeva, 2004) with tidal velocities of 2-4 cm s$^{-1}$

in Herald Canyon, which is much smaller than the residual current velocities. The vertical resolution of the LADCP data is 4

m and the velocities were interpolated onto the 1 m resolution of the CTD data for the volume transport calculations.


Geostrophic shear velocities for both cruises were computed from the CTD data using the Gibbs-SeaWater Oceanographic

Toolbox (McDougall and Barker, 2011). They were then interpolated onto the CTD stations and the bottom triangles filled

assuming constant shear. To calculate absolute geostrophic velocities, we assured that the vertical mean of the geostrophic





shear was zero and referenced to the vertical mean LADCP velocity. For the two outermost stations of each section, we used
the LADCP cross-section velocities.

No LADCP data are available for ISSS-08. Instead, we exploited the fact that wind conditions were very calm over the days
of the ISSS-08 survey with wind speeds below 2.7 m/s during this period (Fig. S1). Consequently, the ship's drift was
predominantly due to surface currents. An estimate of the surface currents was obtained from the ship's GPS positions during
the period over which she was drifting freely at each station. The resulting ocean surface velocities were also de-tided using
AOTIM (Padman and Erofeeva, 2004). The surface current speeds obtained for ISSS-08 (0.1-0.27 m s$^{-1}$ for section 3, which
is common to both surveys) are consistent in magnitude with the SWERUS-C3 LADCP velocities (0.03-0.25 m s$^{-1}$ for surface
speeds, 0.05-0.27 m s$^{-1}$ for vertical mean speeds for section 3). Figure 2 shows that the ISSS-08 surface velocities for section
3 have the same pattern of northward flow in the eastern and southward flow in the western part of Herald Canyon also present
in the 2014 SWERUS-C3 LADCP observations, as well as in the 2004 and 2009 observations presented in Linders et al.
(2017). This gives us confidence in using the surface currents derived from the ship's drift to reference our geostrophic shear
velocities.


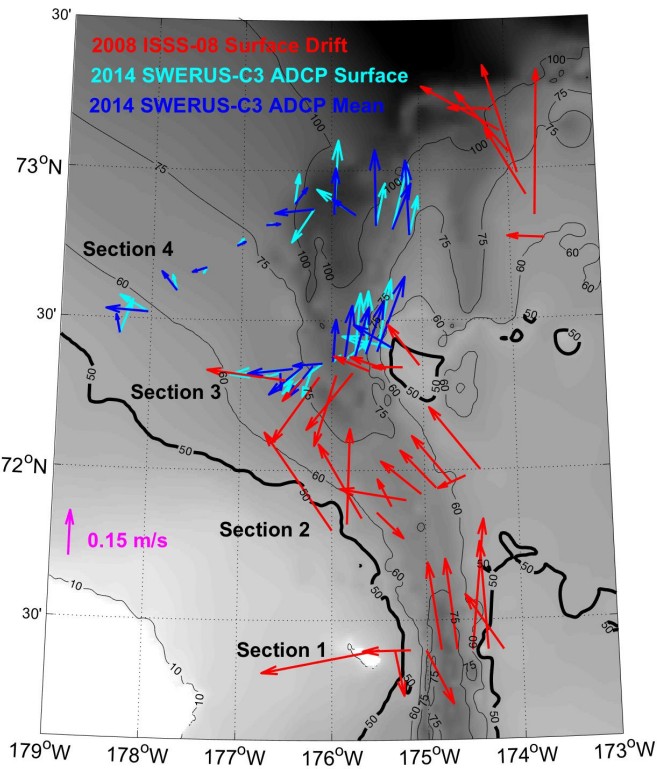

**Figure 2: Circulation pattern in Herald Canyon and reference velocities for the geostrophic shear velocities. Shown are 2014 SWERUS-C3 vertical mean LADCP velocities (blue arrows), 2014 SWERUS-C3 surface LADCP velocities (cyan arrows), and 2008 ISSS-08 surface velocities from ship drift and navigation data (red arrows). Bathymetry (gray-scale shading and black contours) is from the International Bathymetric Chart of the Arctic Ocean (IBCAO, Jakobsson et al., 2012).**

To evaluate the robustness of the flow pattern referencing to either vertical mean or surface currents, we referenced the geostrophic shear for the SWERUS-C3 sections to both surface and vertical mean LADCP velocities (Fig. 2) and compared them to the original ADCP velocities during the 2014 study. Figure S2 shows that the general features of the cross-section velocity fields are consistent between all three, in particular the location of the boundary between northward and southward flow. Volume transports computed using the different velocity fields agree closely (Fig. S3).

**2.1.2. Biogeochemistry**

All chemical measurements discussed in this contribution were performed onboard the research vessels. The samples were stored cold and dark before determination, which was carried out within a day of sampling. In 2008 nutrients (phosphate, nitrate, and silicate) were determined by a SMARTCHEM 200 discrete analyzer (Westco Scientific Instruments/Unity Scientific). The samples were filtered before analysis and evaluated by a 6 to 8-points calibration curve, precision being ~1%. In 2014 the same nutrients were measured onboard using a four-channel continuous flow analyzer (QuAAtro system, SEAL





Analytical) giving a precision between 1 and 3%. The quality was assured by automatic drift control using certified reference materials (CRM) solutions prepared from commercial ampoules (QC RW1, Batch VKI-9-3-0702), except for silicate where no CRM is available. During both cruises oxygen was determined using an automatic Winkler titration system, giving a precision of ~1 µmol kg$^{-1}$. Dissolved Inorganic Carbon (DIC) was determined by a coulometric titration method based on Johnson et al. (1987), having a precision of ~2 µmol kg$^{-1}$, with the accuracy set by calibration against CRM, supplied by A. Dickson, Scripps Institution of Oceanography (USA).


The chemical variables on both cruises were sampled every second station at 8-10 depths per station. To compute transports of nutrients and dissolved inorganic carbon, we interpolated the bottle sample concentrations first vertically onto the 1 m resolution of the CTD data and then horizontally onto the intermediate stations. The achieved fields where then used to compute the average concentrations of the different water masses, as identified by their T/S properties.

**2.2 Model description**

We utilize results from the Regional Arctic System Model (RASM) to assess the circulation and water mass characteristics in the Chukchi Sea. RASM has been developed over the past decade and each component, as well as the fully-coupled system, has been thoroughly evaluated (Brunke et al., 2018; Cassano et al., 2017; Clement Kinney et al., 2020; DuVivier et al., 2016; Hamman et al., 2016, 2017; Jin et al., 2018; Roberts et al., 2015; Roberts et al., 2018). RASM is a high-resolution atmosphere-
ice-ocean-land regional model with a domain encompassing the entire marine cryosphere of the Northern Hemisphere, including the major inflow and outflow pathways, with extensions into the North Pacific and Atlantic oceans. RASM has been developed in order to represent Arctic relevant processes with coupling among model components at high spatio-temporal resolution. It is a fully coupled regional Earth system model with components including, atmosphere, ocean, sea ice, marine biogeochemistry, land hydrology, and a river routing scheme. All the components are coupled using the flux coupler of Craig
et al. (2012). The RASM domain includes all sea-ice covered oceans in the Northern Hemisphere, Arctic river drainage, and large-scale atmospheric weather patterns. The components of RASM are the Weather Research and Forecasting (WRF) atmosphere model, the Variable Infiltration Capacity (VIC) land hydrology model, and the Los Alamos National Laboratory (LANL) Parallel Ocean Program (POP) and Sea Ice (CICE) Models. The model horizontal resolution is 50 km for WRF and VIC, and either 1/12° (9km) or 1/48° (2km) for POP (ocean) and CICE (sea ice).

**3 Results**

In this study we consider the dominating water masses of the Chukchi Sea: Bering Summer Water (BSW), Alaskan Coastal Water (ACW), Siberian Coastal Water (SCW), Winter Water (WW), Summer Water (SW), Melt Water (MW), and Atlantic Water (AW). The temperature and salinity definitions of these water masses follow Linders et al. (2017) and are shown in the





temperature-salinity (T-S) diagrams in Fig. 3. We focus our analysis on the WW and BSW, as they carry the highest nutrient

concentrations, as illustrated by silicate in Fig. 3, and contribute to the halocline of the deep Arctic Ocean. We define Winter

Water (WW) as water with a temperature less than -1℃ and salinity above 31. Bering Summer Water (BSW) is warmer than

WW with a maximum temperature of 3 ℃ and has a salinity of 30-33.3.


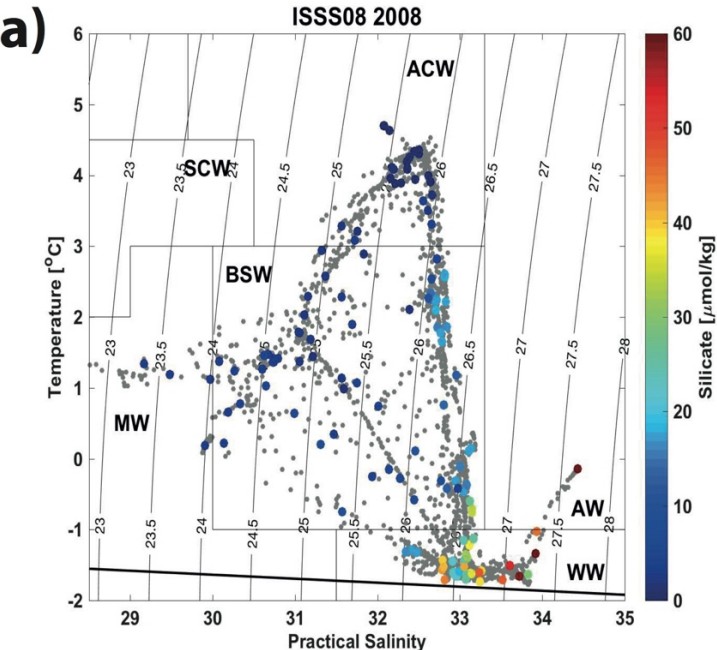

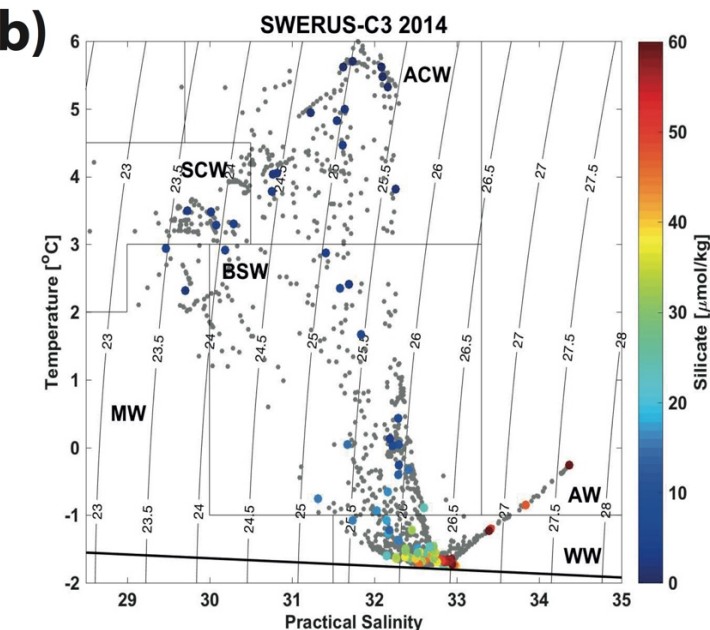

**Figure 3: Temperature-salinity diagrams for sections 1-4 in Herald Canyon (section locations shown in Fig. 1b) for (a) 2008 and (b) 2014. Gray dots are the CTD data at 1 dB depth resolution, coloured dots are from the bottle data colour-coded by silicate concentration. Water masses are defined following Linders et al. (2017): WW = Winter Water, MW = melt water, BSW = Bering Summer Water, AW = Atlantic Water, SCW = Siberian Coastal Water, ACW = Alaskan Coastal Water. The bold black line marks the surface freezing point.**






### 3.1 Observations of hydrography and circulation

Section 3, at about 72° 20′ N in the northern part of Herald Canyon (Figs. 1b and 2), was occupied both in the beginning of September 2008 and the end of August 2014 and was used to compare the water mass characteristics, distribution, and circulation pattern in the two years (Fig. 4). In 2008 WW dominates the western part of the canyon and BSW its eastern part

(Fig. 4e), while WW extends further east in the canyon in 2014 (Fig. 4f). Some of the shallow water with BSW characteristics in the western part of the canyon in both years is probably of local origin when surface water mixes with the underlying WW.





**Figure 4: Temperature (a and b), practical salinity (c and d) for section 3 at approx. 72° 20´ N (Fig. 1b and Fig.2) in 2008 (a, c, e) and 2014 (b, d, f). e) and f) show the spatial distribution of Winter Water (WW, blue) and Bering Summer Water (BSW, red) as defined following Linders et al. (2017) (see also Fig. 3) together with cross-section velocities (m s⁻¹) computed as described in section 2.2. Solid contours denote northward (positive) velocities, while dashed ones represent southward (negative) ones. The white contours in a and b show the temperature structure in the WW. Those in c and d are potential density σ₀ (kg m⁻³).**

When comparing the T-S characteristics of the water masses in Herald Canyon in the 2008 and 2014 observations (Fig. 3), it is apparent that the cold WW had significantly lower salinities in 2014 than in 2008. The mean WW salinity in 2014 was 0.56 lower than in 2008 (Table 1). The freshening in the BSW is weaker (0.09 between 2008 and 2014), and its mean temperature shows a decrease of 0.84°C from 2008 to 2014 (Table 1). This temperature change can partly be explained by the fact that the 2008 section extends further east onto Herald Shoal (Fig. 1) capturing a larger part of the BSW flow and partly by the fact that





the freshening of the WW in 2014 has weakened the density front between BSW and WW leading to enhanced exchange and mixing (Fig. 4c and d).

**Table 1: Mean properties of the Winter Water (WW) and Bering Summer Water (BSW) observed in 2008 and 2014.**

| Water mass | Salinity | Temp | PO$_4$ (µmol L$^{-1}$) | NO$_3$ (µmol L$^{-1}$) | SiO$_2$ (µmol L$^{-1}$) | DIC (µmol kg$^{-1}$) |
|---|---|---|---|---|---|---|
| WW-2008 | 33.22 | -1.55 | 2.46 | 13.5 | 39.1 | 2243 |
| WW-2014 | 32.66 | -1.61 | 1.77 | 11.7 | 40.7 | 2224 |
| BSW-2008 | 32.25 | 1.12 | 1.49 | 5.1 | 11.3 | 2084 |
| BSW-2014 | 32.16 | 0.28 | 1.04 | 4.2 | 10.0 | 2107 |

Net total volume transports across section 3 of 0.279 Sv ($10^6$ m$^3$ s$^{-1}$) southward in 2008 and 0.240 Sv northward in 2014 (Table 2) indicate that flow pattern and transport in this part of Herald Canyon are highly variable. Flow in the canyon at section 3 is pre-dominantly barotropic, southward in the western part of the canyon and northward on its eastern flank (Fig. 4e and f and Fig 5). Both surface and vertical mean ADCP velocities in 2014 and surface velocities from the ship's drift in 2008 show this pattern with stronger southward flow in the western canyon in 2008 that extends eastward to 175.6°W (Fig. 4e). This results

in 0.231 Sv of southward WW transport across section 3 in 2008 with a negligible northward component (Table 2). BSW re-circulates southward in the center of the canyon with a transport of 0.106 Sv southward, and a northward transport of 0.052 Sv at the eastern end of the section (Fig. 4e). In 2014, a westward shift in the boundary between northward and southward flow and the eastward extension of WW (Fig. 4f) results in a weaker southward WW transport of 0.073 Sv and a northward WW transport of 0.127 Sv in the central canyon that contains the core of the coldest WW with temperatures < -1.7 °C (Fig. 4b

and f). BSW flows predominantly north with a net transport of 0.118 Sv (Table 2).





**Table 2: Volume transports computed from observations according to section 2.2 for the sections in Herald Canyon. All transports are in Sv ($10^6$ m$^3$ s$^{-1}$). Negative values denote southward transports, positive northward ones. Note that the values for the total volume transport are larger than the sum of the WW and BSW transports due to the presence of other water masses in Herald Canyon (Fig. 3).**

| Section | 1 | 2 | 3 | 3 | 4 |
|---|---|---|---|---|---|
| Latitude | 71º 25´ N | 71º 55´ N | 72º 20´ N | 72º 25´ N | 72º 40´ N |
| Year | 2008 | 2008 | 2008 | 2014 | 2014 |
| Net Total | 0.474 | 0.329 | -0.279 | 0.240 | 0.618 |
| Total North | 0.572 | 0.390 | 0.082 | 0.357 | 0.745 |
| Total South | -0.098 | -0.061 | -0.361 | -0.117 | -0.127 |
| Net WW | -0.019 | 0.191 | -0.231 | 0.054 | 0.349 |
| WW North | 0.004 | 0.198 | 0.000 | 0.127 | 0.445 |
| WW South | -0.023 | -0.007 | -0.231 | -0.073 | -0.096 |
| Net BSW | 0.293 | 0.051 | -0.054 | 0.118 | 0.082 |
| BSW North | 0.363 | 0.093 | 0.052 | 0.132 | 0.089 |
| BSW South | -0.070 | -0.042 | -0.106 | -0.014 | -0.007 |

In the southern Herald Canyon in 2008, flow across sections 1 and 2 was predominantly directed northward (Fig. 5d and e) with net total volume transports of 0.474 and 0.329 Sv, respectively (Table 2). Linders et al. (2017) present 2009 LADCP observations of northward flow across the whole canyon similar to the 2008 section 2 (Fig. 5d) suggesting that this flow pattern is not unprecedented. Consequently, section 2 has a northward WW transport of 0.198 Sv and little southward WW flow (0.007 Sv). At section 1, very little WW was observed and its flow was thus small. The net northward transport of 0.474 Sv across section 1 is dominated by a northward BSW transport of 0.363 Sv in the eastern part of the canyon (Fig. 5e) with a smaller southward component of 0.070 Sv along its western flank. The reduction of the northward flow of BSW to 0.093 Sv at section 2 and 0.052 Sv across section 3 with a re-circulation of 0.106 Sv in the western trough suggests there may be circulation patterns in the trough that block the transport of BSW towards the Arctic Ocean. Our observations do not, however, allow us to say how often these occur or how persistent they are.




**Figure 5: Cross-section velocities in 2008 for sections from north (b), (d), and (e) and in 2014 (a) and (c). Shown in all panels is**
**260    geostrophic shear referenced to the cross-section component of de-tided surface velocities from ship's drift (2008) and LADCP**
**(2014). The white contours show temperature from -1 to 8 °C at 1 °C intervals, the cyan ones from -1.3 to -1.7 °C at 0.2 °C to highlight**
**the temperature structure within the WW.**

265    In 2014, 0.089 Sv of BSW flows north across the eastern part of section 4 into the Arctic Ocean (Fig. 5a and Table 2), smaller,

but of a similar magnitude to the BSW transport across section 3. WW is found at depths between 20 and 100 m across section

4 and dominates the total transport with a northward flow of 0.445 Sv in the mouth of Herald Canyon and a southward transport

of 0.096 Sv on its western flank onto the Chukchi continental shelf. Both of these WW transports contain cores with

temperatures < -1.7 °C indicating recent winter ventilation (Fig. 5a).




Our sections represent snapshots of a circulation that is predominantly barotropic in a shallow ocean area and therefore is likely strongly influenced by changes in the wind field over synoptic timescales leading to strong variability of both current pattern and strength. Furthermore, the sections do not cover the full width of Herald Canyon and thus water is also flowing outside of these sections. For this reason, we compare the observations with model results where the sections can be selected

in suitable ways and allow us to gain an understanding of the spatial and temporal variability of the flow over a larger area and longer timescales.

## 3.2 Modelled circulation

The numerical modelling results support the general circulation scheme of the Chukchi Sea (e.g. Brugler et al., 2014) but show

a more detailed picture both in time and space.  Mean circulation in the upper 100 m (Fig. 6ab) is similar to the schematic circulation presented in Fig. 1 when averaged over a 10-year period (2006-2015).  Our results show a north-westward flowing Chukchi Slope Current, similar to recent results by Leng et al. (2021), who used a combination of modelling and observations to examine this flow in detail.  Limited observations in Long Strait (Woodgate et al., 2005) suggest a north-westward flow and our model results (Fig. 6) are in agreement.  When we look at the circulation averaged over a summer month when observations

were collected (e.g. September 2008; Fig. 6cd) we see more complex circulation across the shelf, as well as eddies in the Beaufort Gyre.  The circulation in August 2014 (during the second field expedition; Fig. 6ef) shows much higher speeds in the East Siberian Sea and a weaker flow around Wrangel Island than in September 2008.  When comparing the 9-km model output and the 2-km model output there is little difference in the long-term means, but at shorter time scales we tend to see higher speeds and more eddies in the 2-km simulation.







**Figure 6: Upper 100m averaged speed (shading) and velocity vectors (cm/s) from model output. The left column (a, c, e) is from the 9km model results and the right column (b, d, f) is from the 2km model results. The top row (a, b) is a 10-year mean from 2006-2015. The middle row (c, d) is a mean for September 2008. The bottom row (e, f) is a mean for August 2014. Every 4th vector for the 9km and every 16th vector for the 2km is shown. The red line indicates the location of section 3 discussed elsewhere.**





Figure 7 shows the modelled circulation zoomed in on Herald Canyon. The 10-year mean (Fig. 7ab) shows speeds of up to 10 cm/s northward in Herald Canyon. The summer months show stronger anticyclonic flow around Wrangel Island, particularly during September 2008 (Fig. 7cd), with speeds up to 12 cm/s north of Wrangel Island. The modelled circulation also suggests

that in 2014 water flowing northward in eastern Herald Canyon may have been sourced from flows across Herald Shoal (Fig. 7 c-f). This may offer an explanation for higher nutrient concentrations in the BSW in 2014 (Table 1). The largest differences between the 9-km and 2-km simulations are found north of the 100m isobath with more complex circulation in the 2-km simulation.



**Figure 7: Upper 100m averaged speed (shading) and velocity vectors (cm/s) from model output. The left column (a, c, e) is from the 9km model results and the right column (b, d, f) is from the 2km model results. The top row (a, b) is a 10-year mean from 2006-2015. The middle row (c, d) is a mean for September 2008. The bottom row (e, f) is a mean for August 2014. Every 2nd vector for the 9km and every 8th vector for the 2km is shown. The red line indicates the location of section 3 discussed elsewhere.**




### 3.3 Modelled volume flux in Herald Canyon across section 3

Next, we will examine the modelled volume transport during August and September of 2008 and 2014, which encompasses the time period when observations were collected. Time series of daily mean volume transport across section 3 show a large range of variability in the model results (Fig. 8). Over the 2-month period of August-September 2008 (Fig. 8) the net volume transport ranged from -0.4 to 0.4 Sv (negative is southward) with a mean of 0.109 Sv. A flow reversal occurred within a week's time between September 7 and September 14. We show vertical sections of temperature, salinity and velocity on those

two days for a comparison of these extremes (Fig. 9). The velocity core was 0.3 m s$^{-1}$ southward on September 7 and 0.2 m s$^{-1}$ northward on September 14. The model is underestimating the temperature in the upper layers with values only as high as 2 °C, whereas the observed temperature (Fig. 4a) reached 4 °C. The upper layer salinity is slightly higher in the model results than in observations (Fig. 4c).





**Figure 8: Daily modelled volume flux from the 9km simulation across section 3 during August-September 2008 for all water masses (a), BSW only (b), and WW only (c) and during August-September 2014 for all water masses (d), BSW only (e), and WW only (f). The gray box shows the time period during which observations were collected. Inflow values are northward and outflow is southward.**






Over the 2-month period of August-September 2014 the net volume transport ranged between -0.5 Sv to 0.35 Sv with a mean of -0.007 Sv (Fig. 8). There was strong, persistent southward flow during the last 10 days of September 2014, in contrast to the northward flow during the first half of September. Vertical sections of temperature show higher values (up to 3°C) and lower salinity (< 30) in 2014 compared to 2008 (Fig. 10).


The model results allow us to place the observations at or close to events of northward (2008; Fig. 8) and southward (2014) net volume transport of similar magnitude (Table 2). This suggests that the model is able to realistically reproduce the timing and variability of changes in circulation on synoptic timescales discussed above and shown in Figs. 9 and 10. The observed volume transports in both years are larger in magnitude and not in the same direction as the modelled August-September

means. This implies that caution is required when using observations from synoptic, hydrographic sections to estimate the transport of heat, freshwater, nutrients, and carbon through Herald Canyon into the deep basin.



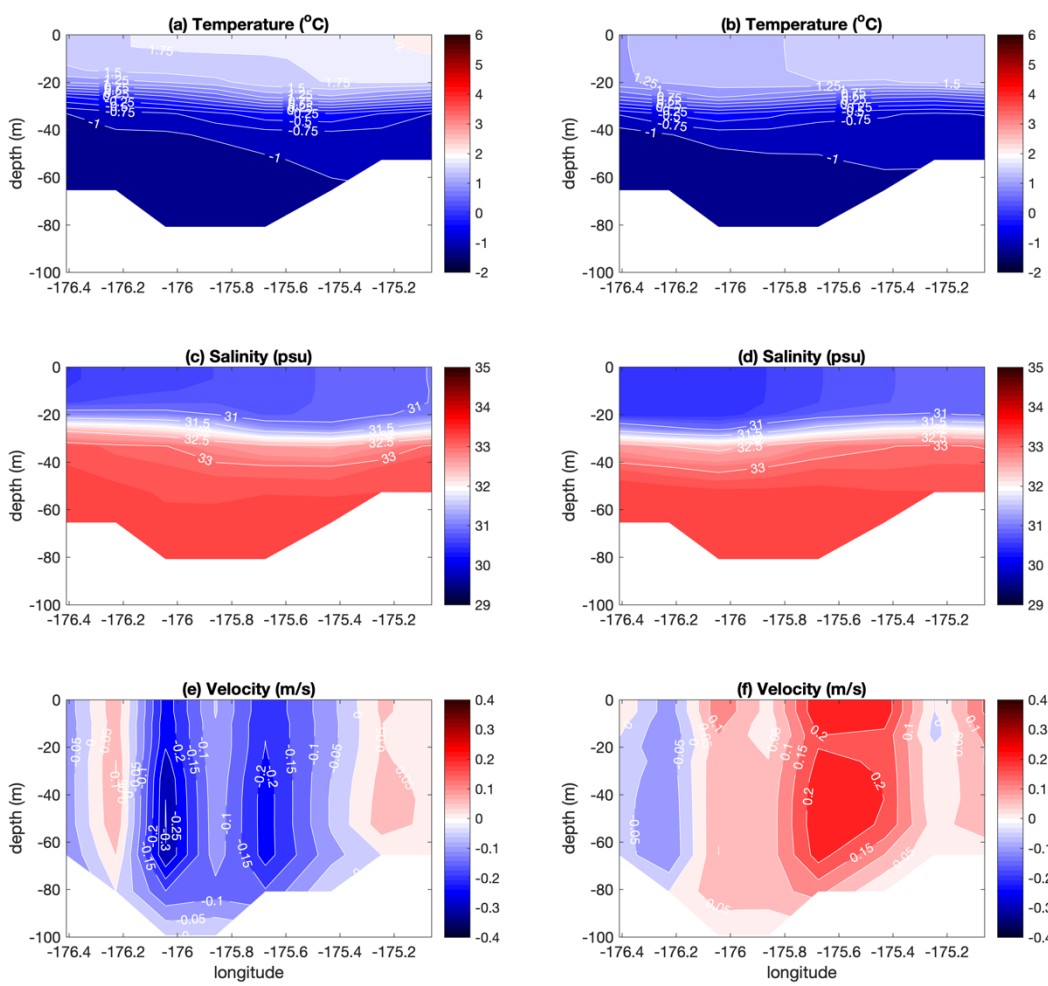

**Figure 9: Modelled vertical sections from the 9km simulation across section 3 on September 7, 2008 (left) and September 14, 2008 (right).**




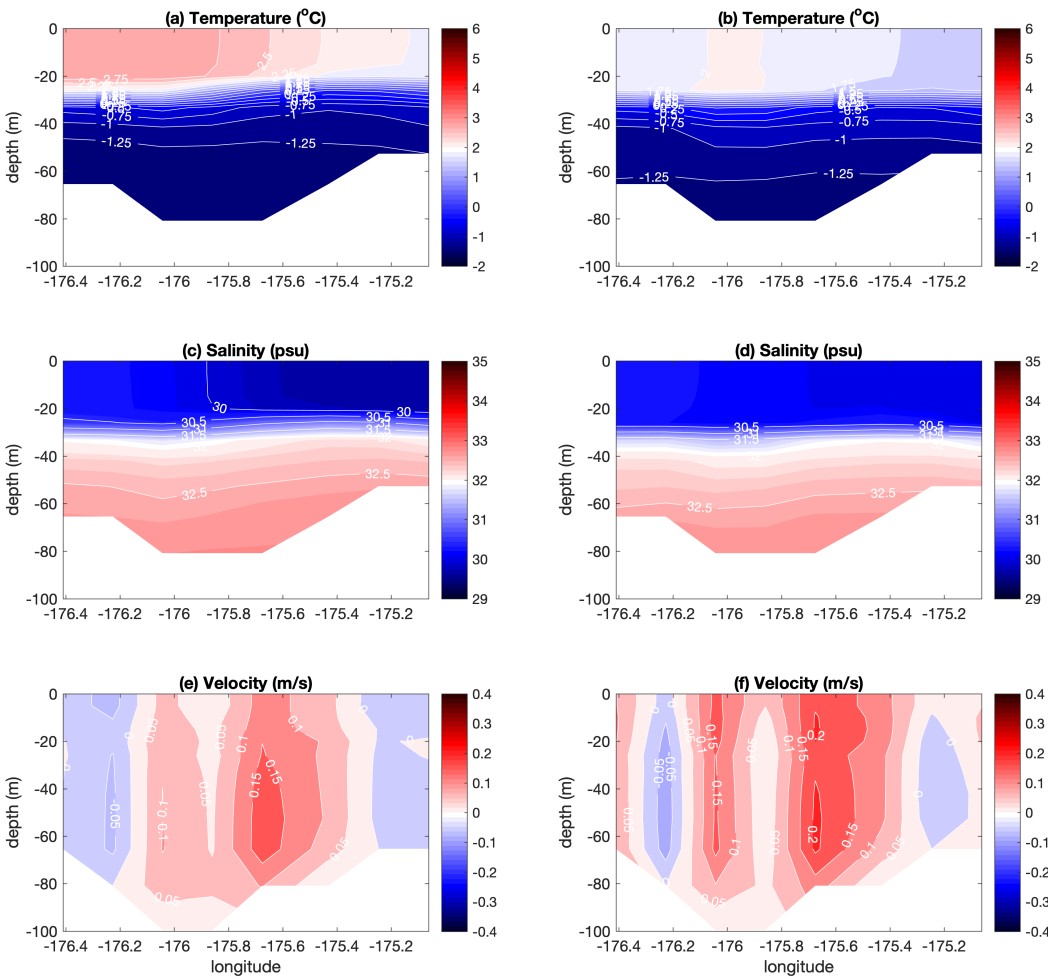

**Figure 10: Modelled vertical sections from the 9km simulation across section 3 on August 27, 2014 (left) and September 11, 2014**
**(right).**

The amount of BSW and WW is similar during the August-September time frames (Figs. 9 and 10) from the model results.
For example, the northward component of the mean flow consists of 0.105 Sv of BSW and 0.091 Sv of WW during 2008. At
the same time, the southward component of the mean flow consists of 0.043 Sv of BSW and 0.045 Sv of WW. There is more
north-south variability in the time series from 2014, as compared to 2008. The lower means reflect this, with the northward
component of the mean flow consisting of 0.056 Sv of BSW and 0.049 Sv of WW in 2014 and the southward component
consisting of 0.063 Sv of BSW and 0.058 Sv of WW.

Figure 11 shows the modelled monthly mean volume transport across section 3 during 1980-2017. This gives a long-term
perspective of the flow through Herald Canyon and indicates that the modelled mean is 0.032 Sv northward for all water




masses combined. This includes the northward (0.063 Sv) and southward (0.031 Sv) components. BSW transport appears to be the strongest in the fall months and has been increasing in prevalence in recent years in agreement with the earlier warming seen in spring in long-term mooring observations in Bering Strait (Woodgate, 2018). The most notable feature in the time series occurred in November 2017, when the model showed a strong, persistent flow reversal in Herald Canyon with southward

speeds over 30 cm/s (Fig. S4) averaged over the upper 100m.

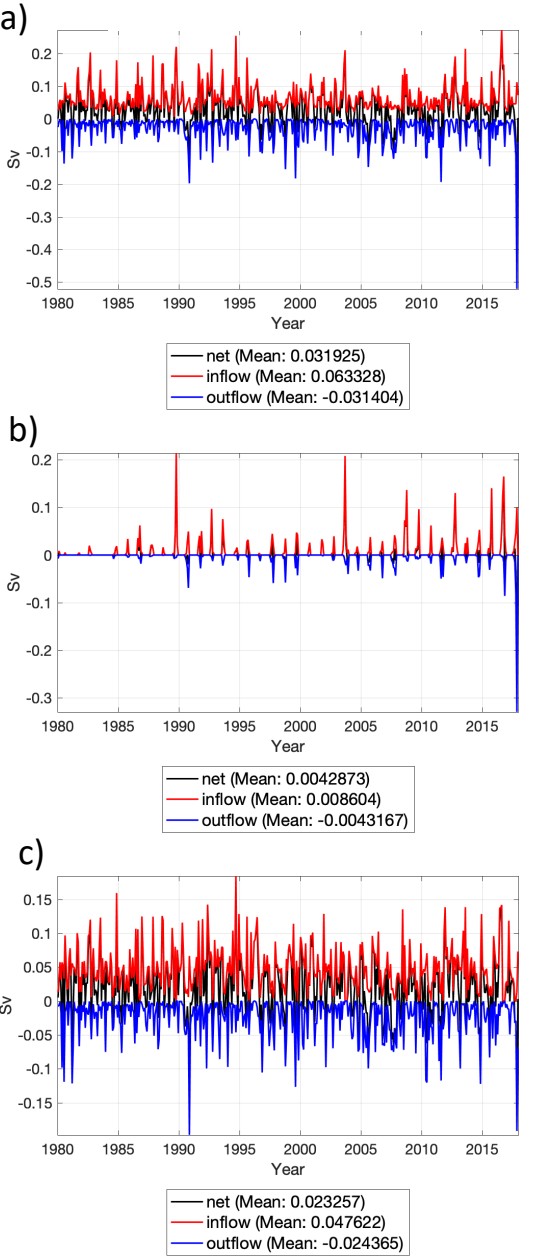

**Figure 11: Monthly mean modelled volume flux (Sv) from the 9km simulation across section 3 during 1980-2017 for all water masses (a), BSW only (b), and WW only (c). Inflow values are northward and outflow is southward.**



## 3.4 Nutrient concentrations and transports

The nutrient and DIC concentrations across Section 3 in 2008 and 2014 show the classical summer distribution of low values in the surface and high values in the deeper layers (Fig. 12). The dominating process for this pattern is primary production in the top 20 - 30 m, followed by sedimentation and mineralization at the sediment surface where the nutrients and DIC are transported back to the bottom water. The highest concentrations of all properties are found more to the west in 2008 than in 2014, consistent with the distribution of WW (Fig. 4ef). Maximum concentrations of nitrate and silicate were similar in 2008 and 2014, while both phosphate and DIC were higher in 2008.







**Figure 12: Observed nutrient and DIC sections at latitude about 72° 20′ N (noted as 3 in Fig 1b) in 2008 (a, c, e, g) and 2014 (b, d, f, h).**


To compute the transports, the mean concentrations of nutrients and DIC were calculated by first interpolating the bottle sample concentrations vertically onto the T-S points from the CTD reading of each station where they were available and, subsequently, horizontally onto the stations in between. Using this concentration field, the mean concentration for each watermass was computed for the T-S ranges shown in Fig. 3, which avoids mean concentrations being dependent on the

number of measurements. The results show substantial differences in phosphate between the years, some differences in nitrate, and fairly constant silicate concentrations (Table 1). DIC appears to be quite constant, but one has to consider that its concentration is strongly salinity-dependent. When the WW concentrations are corrected for the salinity difference, the 2014




DIC concentration becomes 20 µmol kg⁻¹ higher than in 2008 and for the BSW the concentration becomes 31 µmol kg⁻¹ higher. When these differences in concentration are multiplied with the classical Redfield-Ketchum-Richards ratios 1:16:106 for

P:N:C (Redfield et al., 1963), they correspond to a shift in phosphate of 0.19 and 0.30 µmol L⁻¹ for WW and BSW, respectively, and in nitrate of 3.0 and 4.7 µmol L⁻¹ for WW and BSW, respectively. These are significant differences relative to those observed in phosphate and nitrate. Utilizing the mean concentrations of Table 1 together with the volume transport of Table 2, net property transport across the different sections and years are computed (Table 3). Using the 37-year mean modeled volume transports of Fig. 11, the transports of Table 3 would roughly halve for the WW and more or less vanish for BSW.

**Table 3: Transport of nutrients and DIC on a daily basis based on the measured data for the different sections, units are in 10⁸ mole day⁻¹. Volume transports are the total from Table 2.**

|          | Year    | 2008  | 2008 | 2008  | 2014 | 2014 |
|----------|---------|-------|------|-------|------|------|
|          | Section | 1     | 2    | 3     | 3    | 4    |
| Silicate | WW      | -0.6  | 6.5  | -7.8  | 1.9  | 12.3 |
|          | BSW     | 2.9   | 0.5  | -0.5  | 1.0  | 0.7  |
| Nitrate  | WW      | -0.2  | 2.2  | -2.7  | 0.5  | 3.5  |
|          | BSW     | 1.3   | 0.2  | -0.2  | 0.4  | 0.3  |
| Phosphate| WW      | -0.04 | 0.4  | -0.5  | 0.08 | 0.5  |
|          | BSW     | 0.4   | 0.07 | -0.07 | 0.1  | 0.07 |
| DIC      | WW      | -37   | 370  | -448  | 104  | 671  |
|          | BSW     | 528   | 92   | -97   | 215  | 149  |

## 4 Discussion

WW in 2014 was significantly fresher than in 2008. However, the larger volume of this watermass present in Herald Canyon

in 2014 led to a further eastward extent compared to 2008 (Figs. 4cd, 11cd). There was an extreme sea ice minimum in 2007, especially in the East Siberian and Chukchi Seas (Comiso et al., 2008). The sea ice cover re-formed more slowly in October-December 2007 than in October-December 2013 with weaker heat loss from ocean to atmosphere in autumn 2007 (Fig. 13). Winds over the Chukchi Sea are generally northeasterly in winter, piling water against the Siberian coast and preventing its northward flow until winds weaken in spring (Pickart et al., 2010). During autumn and early winter 2013/14 winds were more

westerly than in 2007/08 tending to drive water off the East Siberian shelf (Fig. 13). The resulting reduction in residence time could potentially reduce the salinity enhancement of the shelf waters due to brine release during sea ice formation, leading to a larger volume of fresher WW being formed in winter 2013/14. Despite weaker heat loss in winter 2007/08, the longer residence time on the East Siberian shelf allowed the formation of a smaller volume of more saline WW. In addition, the large-


scale freshening of the Arctic Ocean over recent decades, particularly in the Canada Basin (Haines et al., 2015) may have

contributed to the difference in WW salinities between 2008 and 2014. The observed (and modelled) freshening of the WW

in 2014 reduces its density sufficiently to weaken the density front between WW and BSW across Herald Canyon potentially

enhancing the exchange between the water masses (Figs. 4, 7).

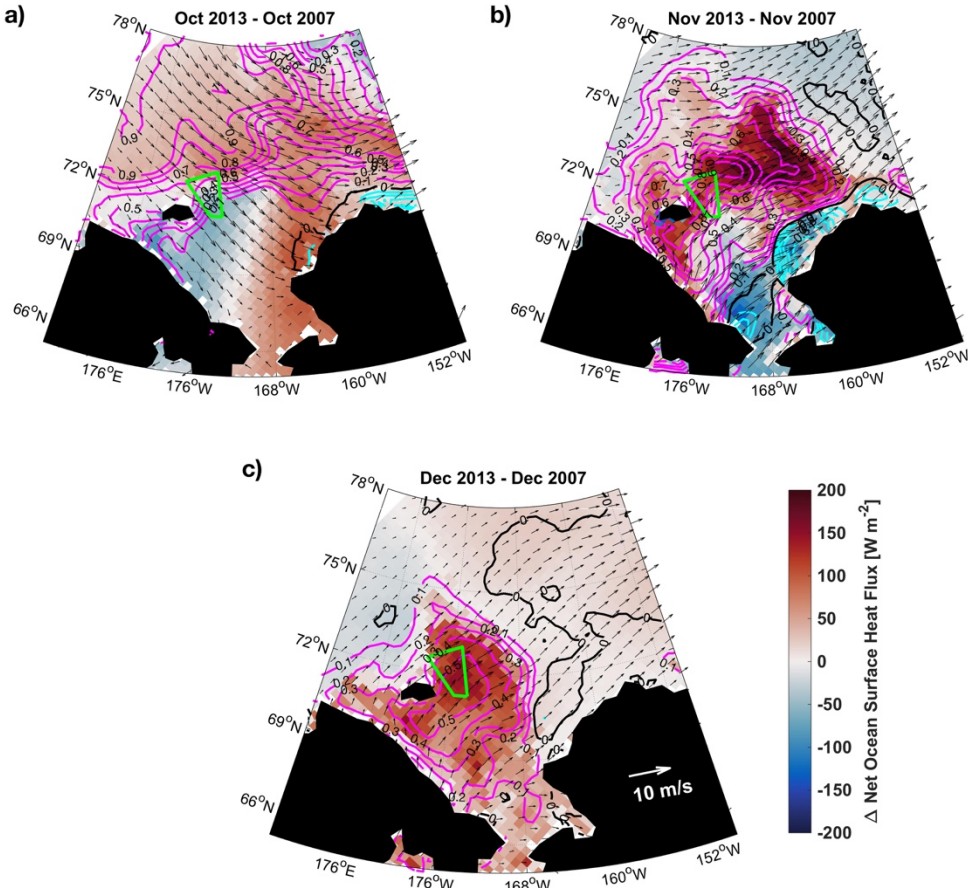

**Figure 13: Difference in wind velocities (black vectors), sea ice concentration (contours, magenta positive anomalies, i.e. more sea**
**ice in 2013, cyan negative anomalies), and net ocean surface heat flux (shaded) for the autumn and winter sea ice growth season**
**preceding the two cruises. Heat fluxes are defined positive upward, i.e., positive anomalies correspond to a larger heat loss by the**
**ocean in 2013. Monthly means for a) October 2013 – October 2007, b) November 2013 – November 2007, c) December 2013 –**
**December 2007.**

Modelled volume transport showed a large range of variability in Herald Canyon on a daily timescale (Fig. 8). Flow reversals

were common and occurred during August-September of both 2008 and 2014. This is consistent with observational data from

1990-1991 in Herald Canyon, which showed flow reversals lasting a week or more that were correlated with the wind field

(Woodgate et al., 2005). Synoptic storms enter the Chukchi Sea frequently during the late summer/fall time period and are a

driving force for the flow in Herald Canyon (Pickart et al., 2010). The ocean component of RASM is coupled to a relatively





high-resolution atmospheric component within the system model, however errors in the representation of atmospheric fields
would likely be in the direction of underrepresentation of the strength of storms. This leads us to conclude that the large
variability in modelled flow is valid and that short-term measurements of the speed and direction of the flow are not necessarily
representative of the mean over time.

The currents computed from the observations, as well as the numerical modeling results, show southward flowing water in the
western Herald Canyon (Figs. 4 and 7). This southward-flowing water most likely has the East Siberian Sea as a source which
is supported by the hydrographic signature of the water as well as the general circulation from the model results (Fig. 7). It has
high nutrient concentrations, typical for decay of organic matter, but not high enough salinity to come from any substantial
depth along the continental margin (Fig. 14a). Also, its temperatures are close to the freezing point, significantly lower than
the temperature of water at the shelf break with the same high silicate concentration (Fig. 14b). Furthermore, it has a higher
oxygen concentration than the slope water at the high silicate concentration (>40 μmol L$^{-1}$), as exemplified by the 2014 data
in Figure 14c. The only exception is the deepest water in the northern section, with silicate concentrations above 70 μmol L$^{-1}$;
water that most likely includes some intrusions of the slope current as it also has higher salinity and temperature compared to
the WW.



Figure 14. Plots of SWERUS-C3 East Siberian Sea shelf break data compared with those of Herald Canyon: Silicate versus salinity (a), silicate versus temperature (b) and oxygen versus silicate (c). Locations of the stations noted in the map of Fig. 1b.

Winsor and Bjork (2000) used a polynya model driven by atmospheric forcing to compute ice, salt, and dense water production in different regions of the Arctic Ocean over 39 winter seasons from 1958 to 1997. The area west of Wrangel Island was one region where many polynyas formed and created high salinity waters from brine produced by ice formation.





Another aspect is the fate of the southward-flowing water to the western Herald Canyon; is it recirculating within Herald Canyon or is it flowing around Wrangel Island back into the East Siberian Sea? Our observational data can not reveal this, but as most of the high nutrient water is deeper than 50 m, a depth that to our knowledge exceeds that south of Wrangel Island,

the most plausible conclusion is a recirculation in the canyon. Also, there are some indications of two cores of high nutrient water in 2014 at section 3, one in the west on the shallower part and one in the deep central canyon (Fig. 12 b, d, f), that could be a result of one south and one north flowing core.

When assuming that the southward flowing water recirculates in Herald Canyon it also means that the net flux gives the total

transport within the geographic range of the section, i.e. if the water that recirculates is included in the north-flowing current. Hence the transports of Table 3 represent snapshots of the supply to the north. It should be noted that this section does not cover all waters that flow either to the south or to the north and thus comes with substantial uncertainties. However, Fig. 5a suggests that more of the northward flow is missed, which indicate that the transports are underestimated. The supply of nutrients contributing to new primary production in the central Arctic Ocean in 2014 can be computed from the transport at

section 4 (Table 3), the most northerly and thus the one that might best represent the flux into the deep basins. Adding together the transport in the BSW and WW we achieve a potential new primary production of $0.9 \times 10^{12}$ mol C yr$^{-1}$ ($11 \times 10^{12}$ gC yr$^{-1}$) using the nitrate flux and $2.4 \times 10^{12}$ mol C yr$^{-1}$ ($28 \times 10^{12}$ gC yr$^{-1}$) using the phosphate flux, all based on the RKR ratios. However, these numbers have to be taken with care as the model annual average volume transport of BSW and WW in section 3 only was 15% of the computed from the measured properties. It is evident, though, that nitrate is the limiting element and that a

volume transport of 0.431 Sv (Table 1) makes a major contribution to the fueling of primary production in this region of the Arctic Ocean. For instance, Anderson et al. (2003) used the phosphate deficit in the central Arctic Ocean to compute an export production that corresponded to $2.5 \times 10^{12}$ g C yr$^{-1}$ of the Canadian Basin photic zone, i.e. the supply by the BSW and WW could contribute 5 times this export if it reached the photic zone. However, this is not expected, at least not under the present oceanographic conditions, but points to the importance of the transport through the Herald Canyon as a source of nutrients to

the central Arctic Ocean. High values of primary production were also modelled in Long Strait and much of the East Siberian Sea during June 2011 (Clement Kinney et al., 2020), apparently driven by a northwestward flow through Long Strait. The arrival of high-nutrient water from the western side of Bering Strait when combined with accelerated early sea ice melt created conditions that were favorable to primary production in the East Siberian Sea.

## 5 Conclusions

Numerical modelling results show that a substantial fraction of Pacific Water that enters the Chukchi Sea through the Bering Strait continues into the East Siberian Sea via Long Strait south of Wrangel Island. Both the model results and hydrographic conditions support that some of this water flows north of Wrangel Island back into the Chukchi Sea and is entrained in the



northward-flowing water of Herald Canyon that supplies the halocline of the central Arctic Ocean. The net transport in Herald Canyon, at a section located about 72º 20′ N, was computed based on observations in September 2008 to be about 0.3 Sv to

the South, of which ~80% had WW properties, while ~20% was BSW. In August 2014, on the other hand, corresponding computations gave a transport to the North of 0.24 Sv of which ~20% was WW and ~50% was BSW, with the remaining ~30% being surface waters. This large variability in transport, including a change in the direction, points to the flow in Herald Canyon being highly variable in time and space, a result of the shallow water environment being sensitive to the wind and sea ice conditions. This is also illustrated by the simulated daily modelled net volume transport for the same section that ranged

from -0.4 to 0.4 Sv (negative is southward) with a mean of 0.109 Sv over the 2-month period of August-September 2008, while it ranged between -0.5 Sv to 0.35 Sv with a mean of -0.007 Sv for the same months in 2014. A long-term perspective for the period 1980-2017 indicates a modelled mean transport of 0.032 Sv northward for all water masses combined, which includes a northward component of 0.063 Sv and a southward of 0.031 Sv. However, the maximum monthly transports are up to 0.2 Sv to the north and nearly as much to the south. The BSW transport has a relatively evident seasonal signal with the strongest

transport in the fall months, also with a tendency for an increasing transport during the last decade. The WW transport on the other hand does not show any seasonal pattern.

The nutrient concentrations in the WW were on the order of 2 µmol L$^{-1}$ for phosphate and 12 µmol L$^{-1}$ for nitrate, with lower values observed in the BSW. Thus, Herald Canyon is a substantial conveyor of nutrients to the Arctic Ocean and using the

volume transport computed from measurements at the most northern section it could support a new primary production of $0.9 \times 10^{12}$ mol C yr$^{-1}$, based on the nitrate flux that is the limiting element. Such a new primary production is more than what has been reported for the Canadian Basin, but one should note that not all of the nutrients in the WW will enter the photic zone in a short time-perspective, and also the measured water volume transport is only 15% of the average modelled. Nevertheless, the nutrient supply through the Herald Canyon is important for primary production and appears to be a source for under-ice

phytoplankton blooms that have been observed (Arrigo et al., 2012) and modelled (Clement Kinney et al., 2020) within the canyon and to the north. This investigation shows the highly dynamic conditions in the western Chukchi Sea and pinpoints a need for long-term monitoring of transport and biochemical properties in Herald Canyon, in order to better understand its role in supply to the central Arctic Ocean and to validate model results.

**Data availability**

CTD data can be accessed at the World Data Center PANGAEA. Chemical and Bathymetric data can be accessed at Bolin Centre Data Base, Stockholm University http://bolin.su.se/data/. The daily 10 m NCEP North American Regional Reanalysis NARR winds and monthly mean surface heat fluxes were downloaded from ftp://ftp.cdc.noaa.gov/Datasets/NARR/Dailies/monolevel/.





## Acknowledgements


The science was financially supported by: US National Science Foundation (Grant Number: GEO/PLR ARCSS IAA#1417888), the Department of Energy (DOE) Regional and Global Model Analysis (RGMA), the Swedish Research Council Formas (contract no 2018-01398), and the Swedish Research Council (contract no 621-2006-3240, 621-2010-4084, and 2012-1680). This work was carried out with logistic support from the Knut and Alice Wallenberg Foundation and from

Swedish Polar Research Secretariat. The Department of Defense (DOD) High Performance Computer Modernization Program (HPCMP) provided computer resources. We thank Steve Okkonen for providing a helpful review of an early version of this manuscript.

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
