# Peer review of "On the circulation, water mass distribution, and nutrient concentrations of the western Chukchi Sea"

_Ocean Science, 2021_

## Author Comment (AC1)

**CC1: Rebecca Woodgate**

Very interested to see this. A few quick questions:

1) You report that cold WW in Herald Canyon is fresher in 2014 than 2008 and discuss (lines ~ 400) that that may be due to changes in advection and brine release. However. I wonder how much of that freshening can be related to the freshening in the Bering Strait, Woodgate and Peralfa-Ferriz, 2021. See, e.g., Figures 5 and 6a of Woodgate, 2018, and data available at psc.apl.washington.edu/BeringStrait.html (monthly mean salinities for example) where you can clearly see 2014 is mostly fresher than 2008. Do you see this freshening in the Bering Strait in your model? And can you quantify how that affects the salinity in Herald Canyon?

Thank you for bringing this article to our attention. Your Bering Strait results show that there is a freshening of about 0.3 between 2008 and 2014 from upstream. This is about half of the WW freshening we observe in Herald Canyon. We have updated the paragraph to add this. However, we have retained our line of argument that changes in circulation and brine release the previous winter on the East Siberian shelf likely also contribute. We have edited the relevant figure to show that 2008 follows the wind pattern identified by Pickart et al. (2010) with strong northeasterlies in late autumn and early winter and a weakening in spring. The 2013/14 winds are very weak both in winter and spring and will not have helped to constrain waters to the shelf west of Herald Canyon.

2) Line 470 .. "a substantial fraction of PW ... continues to the ESS via Long Strait". It would be helpful to be specific as to how much.

We can specify how much in the revision.

3) Related, it would also be interesting to quantify in the model how much of the Chukchi outflow enters the Arctic via Herald Canyon directly (rather than through Barrow Canyon).

We can also quantify how much of the Chukchi outflow enters Herald Canyon directly in the revision.

---

## Author Comment (AC2)

**RC1: REVIEWER 1**

We thank the reviewer for taking time to provide helpful comments that will improve our manuscript. We have provided responses to these comments below in blue color.

This paper presented the structures of the hydrography, velocity and nutrient concentrations in vicinity of the Herald Canyon using the data from two cruises. The similarities and discrepancies of these two cruises were also shown in the model. The model reveals the transports through the Herald Canyon have large variations daily and interannually. As the measurements are quite valuable, from the model I expected to learn more about the mechanisms of the circulation and how it distributes the water masses and biogeochemical properties, which I thought is also the main goal of the study. Unfortunately, the authors put more efforts on describing the model outputs, without diagnosing the mechanisms in detail. The paper mentioned several times that the wind forcing plays an important role in modulating the circulation. The southward-flowing water likely originates from the East Siberian Sea and might recirculates within the Herald Canyon. All of these are less convincing by just characterizing, while I think might be more enlightening if they take the further step of the model. In any case, before presenting the model results, I think the authors need to show the readers how robust the model is by comparing the outputs with the observations. I am not saying everything has to match with each other, but at least show the transports during the two cruises are somehow comparable.

Overall the paper is well-written and clear, I recommend a major revision.

We plan to address your comments by looking more closely at the wind field and providing additional analysis from the model output. This should provide more information on the mechanisms controlling the circulation and biogeochemical property distribution. As you mention below, we have already compared the transports and vertical sections of T, S, and velocity from the observations and model output in Figs. 4, 5, 8, 9, 10 and Tab. 2.

The specific comments and questions:

As the focus of the paper, the conditions of hydrography and circulation in the Chukchi Sea certainly deserve more descriptions in the introduction, instead of the briefly summary in one paragraph. It is also necessary to mention the particularity/importance of the western Chukchi Sea in the introduction.

We have expanded the introduction as you suggest.

Line 65. These water masses are Pacific Summer Water.

Corrected.

Line 67. It should be Bering Summer Water in Pisareva et al. (2015), while was named as Bering Sea Water by Coachman et al. (1975). You would need to adjust your references.

Corrected.

Line 75. Any reference?

Linders et al. 2017 reference added.

Line 165. The model configuration needs more details, i.e. what are the initial conditions and the forcing? How long did the model run?

We will add more details to the model description, such as: The RASM historical (1979-2018) simulation results analyzed here were produced after a 78-year spinup, which started with no sea-ice and the Polar Science Center Hydrographic Climatology (PHC) 3.0 climatological ocean temperature and salinity at rest and was forced with the Coordinated Ocean-sea ice Reference Experiments version 2 (CORE2) reanalysis.

Lines 181-183. Some of the water masses are not appreciate (or not seen in previous studies including the Linders et al. (2017) mentioned in the paper). Is the Winter Water actually the Pacific Winter Water? What about the River runoff? I am not sure what the Summer Water represents for, maybe Pacific Summer Water? But the authors already listed the two types of Pacific Summer Water, Bering Summer Water and Alaskan Coastal Water.

Thank you. We have corrected the water mass labels. We now are consistent with Linders et al. (2017), except that our Pacific Winter Water is a combination of their Remnant Pacific Winter Water and Newly Ventilated Pacific Winter Water.

Figure 3. I am curious why not show the nitrate. I thought biologists care more about nitrate.

The strongest signal in the upper halocline of the Canada Basin is in silicate, and thus it is the most obvious tracer.  Any of the nutrients could be used, but nitrate is the least suitable as it is lost through denitrification in the upper sediment.  As this figure only aims at showing which water masses have the highest nutrients, we choose to use the one with the strongest signal.

Lines 199-200. There must be a reason to make such a speculation. I understand that the authors want to focus on the BSW and WW, but showing all of the water masses in the Figs 4e, 4f may help interpret the water mass mixing that they pointed out.

We have added the surface water masses to panels e and f in Fig 4 and changed the sentence to the following:

"With salinities of 30-32 some of the shallower water with BSW characteristics in the western part of the canyon (Fig. 4) in both years lies on the mixing line between WW and MWR in 2008 and WW and SWC in 2014 (Fig. 3). Being distinctly fresher than the BSW modes that Linders et al. (2017) identify as having a Bering Strait origin and is probably of local origin."

Lines 223-224. This sentence is confusing. Did you mean Fig. 5b?

We have changed the figure reference to Fig. 2.

Line 272. The authors said that the wind is important, but never compared the wind condition between the two cruises.

To address this comment and your comment regarding the discussion of the atmospheric and sea ice variability, we have changed Fig. 13 to include panels that show the surface winds for the week before and including the surveys.
We have added the following to the results section:
'Winds were southerly in the week before and including the 2014 survey (Fig. 13b) and may have enhanced the flow forced by the forward pressure coming from Bering Strait, while strong easterlies in 2008 (Fig. 13a) may have caused a build-up of water towards Wrangel Island that potentially induced stronger southward barotropic flow across section 3. '

Figure 6. Whether the Beaufort Shelfbreak Jet was not well resolved or was just not shown in the subsampled plot. The authors did the comparison of 9km- and 2 km-res model outputs, but never explained why they eventually used the 9km-res data for the following analysis.

The 9km model output was available on a daily timescale (instead of just monthly mean) and so was better for comparing with the observational estimates.

Figure 8. I did not expect to see a good agreement of T/S/U sections between observations and model, but it will be more convincing if you can at least compare the mean transport during each of the occupations.

The mean model transport (and variability) during each occupation is stated in section 3.3 for comparison with the observations.

Line 331. I think the authors cannot make such a statement by comparing the 2-month mean transports in Fig. 8 with the snapshot observations. See my suggestion above.

We will make the comparison more clear and precise in the revision.

Line 346. I see now they present the transports from the model for each occupation, which I think needs to be shown above as a model robustness check before getting into the details.

We can make reference to this earlier in the text.

Lines 355-360. I believe the BSW transport peaks in fall, but it is hardly seen in the Figure 11. I am confused why they discussed the seasonality based on the interannual variation in Figure 11. There are certainly more to discuss in terms of the interannual variation, i.e. any increasing trend in transport as the Bering Strait inflow (Woodgate, 2018).

We will address the increasing trend in transport through Bering Strait in the revision.

Lines 381-382. This sentence is not obvious to me. Does it mean that the comparable DIC concentrations are due to the similar salinity?

The text has been expanded to make our arguments clearer.

Line 395. Fig 11cd?

The reference to 11cd is likely a remnant from an earlier manuscript version. The sentence and reference were removed as part of the revision process.

In the discussion section, how do you define the net ocean surface heat flux? Is it the air-sea heat flux? As shown in Fig.13, the ice covers in the winter, how did you (or model) deal with the ice for the heat flux calculation? If it is the air-sea heat flux, it supposes to be zero in the region where was covered by ice. It seems not the case in this paper (Fig. 13). Did you consider any effect of the advection? The WW may not be locally formed, but be advected from the upstream. The authors argued that the origination and fate of the southward-flow water in the Herald Strait with the limited observations. Why not look at the model output which may provide some evidences? For example, tracking a tracer in the model.

We have removed the heat fluxed from the discussion and figure. You are right in saying that heat flux products in sea ice covered areas are difficult to use and interpret. What we were showing was strictly a "net surface heat flux" not a "net ocean surface heat flux".

We have integrated a very helpful comment from Rebecca Woodgate into this discussion. Her Bering Strait mooring record show that around half of the WW freshening that we observe between 2008 and 2014 does indeed come from upstream.

However, this implies that half of the freshening needs a different explanation and the change in wind pattern between 2007/08 and 2013/14 together with the Pickart et al. (2010) mechanism offers this. We have edited Fig. 13 to show the change in the winds more clearly and intuitively.

---

## Author Comment (AC3)

**RC2: REVIEWER 2**

We thank the reviewer for taking time to provide helpful comments that will improve our manuscript. We have provided responses to these comments below in blue color.

This paper scrutinizes the observations made in 2008 and 2014 (especially in Herald Canyon) and attempts to explain the behavior of WW, BSW, nutrients, etc. using numerical models. The numerical model has been developed and improved vigorously by the authors, and I believe that it is one of the most reliable models applied to the Arctic Ocean. However, even though the model output is provided, it is no different from the description of the observation results, and the details of the mechanism are not mentioned. It is recommended that the paper be improved by examining the results of the numerical model more closely.

~Major points~

Line 227 (Westward shift in boundary between northward & southward flow) : Why did the westward shift of the boundary eventually occur, the mechanism would need to be explained since it affects the flow rate of WW. Line 271-272 says that it is strongly affected by wind stress because of forward pressure. The cross sections of the 9-km resolution model (Figs. 10 and 11) do not show a westward shift. Isn't it necessary to show the wind stress field (field and model) during the observation period? In Fig. 4, the WW seems to be constrained by the topography. What are the results of the 2-km resolution model?

In response to this comment and a comment by reviewer 1 we have changed Fig 13 to include panels that show the winds just before and during the two surveys. We have added the following to the observational part of the results section:

"Winds were southerly in the week before and including the 2014 survey (Fig. 13b) and may have enhanced the flow forced by the forward pressure coming from Bering Strait, while strong easterlies in 2008 (Fig. 13a) may have caused a build-up of water towards Wrangel Island that potentially induced stronger southward barotropic flow across section 3."

Figs 6 & 7: Are you using the 2-km resolution model output only to explain the faster flow speed, the greater number of eddies in the ocean basin and more complex circulation north of 100m isobath? First of all, the authors should add the 100m isobath (there may be one, but I can't see it.). If the average velocity field or

cross-sectional view does not change the results much, then I think only 2-km is sufficient. "source from flows across Herald Shoal" can be said for 2008, can't it?

The differences, as well as similarities, between the 2km and 9km circulation are interesting to modelers and we would like to keep these figures for that reason.

Figs. 8 & 9: The model output of T, S, and velocity shows a fundamentally different structure from the observation: in 2008, the WW is unevenly distributed to the west in the observation, but not in the model output. The structure of the surface layer (up to 20 m depth) is also completely different in 2014. Why is the northward velocity distribution split into two in the model?

We plan to address this comment in the revision.

Discussion: The authors mention heat loss and residence time to explain the fresh WW in 2014. However, these explanations are only speculations at present. Since the model output is available, heat loss and residence time (and impact of brine rejection) can be calculated explicitly by tracer experiments. It should also be possible to study in detail the water mass properties and their sources of variation in upstream and downstream areas with the model output. My personal impression is that WW freshening cannot be explained by local phenomena alone.

We have added a reference to Woodgate and Peralta-Ferriz (2021) who show that the Bering Strait inflow freshened by around 0.3 between 2008 and 2014 explaining around half of the observed WW freshening. We have retained the variability of the winter circulation and sea ice formation that transform the Bering Strait inflow on the East Siberian shelf as an explanation for the remainder of the freshening signal in Herald Canyon. However, we have removed the heat flux discussion following a comment by Reviewer 1.

Since data assimilation is not applied, when explaining the reproducibility of the model, etc., the snapshot of the model output will naturally show some differences from reality. For example, how about using the Ensemble mean of the results from a year with a north wind and a year with a west wind to illustrate how much the velocity structure and WW flow rate changes with wind stress?

We can address the question with more model results in the revision.

~Minor points~

Figs 4. 5, 9 & 10: Please improve the diagram so that we can see the distances between the points.

We can change the horizontal scale from longitude to km in these figures for the revision.

Line 93: The observation period of SMMR is 1978-1987, so it must be SSMI.

Corrected.

Line 109: Isn't the frequency of RDI ADCP 300 "kHz"?

Corrected.

---

## Author Response (AR2)

Response to Reviewer #2

I appreciate the efforts of the authors in making corrections to my comments. However, the paper is still limited to the description of observations and model results, and although the wind field data has been expanded, I think the mechanism is not fully explained.
The question is not only the time variability, but also why there is a flow towards the Long Strait in the first place.
The authors state its existence based on the literature showing observational data, but it is not clear why the flow deviates from the topographic constraints and heads for the Siberian continental shelf. Could it at least explain the factors of this flow shown in the numerical simulation results?

Compared to the mean flow field in Figure 7, the two eastern branches (i.e., the ACC , and current flowing eastern Herald Shoal) seem to have smaller velocities (or transport) in 2008 and 2014. Isn't it possible that if the transport through Barrow Canyon and eastern side of Herald Shoal are small (e.g., due to prevailing northeasterly winds), the Pacific-origin water will have nowhere to go and will head for Long Strait?

Also, in 2008, the clockwise circulation around Wrangel Island was enhanced, but why does this happen? Isn't it because the winds around Wrangel Island are weaker (according to Figure 6) and the topographic constraint becomes more dominant in 2008?

Why there is a flow towards Long Strait and why there is a flow bypassing Wrangel Island is also important for considering future nutrient fluxes to the Arctic Ocean, and I think that because of the lack of observational data, the authors can identify possible dominant factors using their model that could reproduces this flow.

Thanks again to the reviewer for input on our manuscript.  We have added the following text to the Discussion.

The flow through Bering Strait and downstream in the Chukchi Sea has been commonly attributed to forcing by the local winds and a far-field difference in sea surface height (SSH) between the Pacific and Arctic, or so called 'pressure head' (Coachman and Aagaard, 1966; Stigebrandt, 1984). A more recent study (Peralta-Ferriz and Woodgate, 2017) based on the GRACE Ocean Mass Satellite Data and *in situ* mooring data suggests the dominant role of the East Siberian Sea (ESS) in driving the flow through Bering Strait. Their analysis shows westward winds driving northward Ekman transport of shelf waters into the basin, which lowers SSH in the ESS and amplifies the pressure head forcing. While the measurements are very limited, the northwestward flow toward Long Strait originates in the western side of Bering Strait and is most likely (Woodgate et al. 2005) steered toward Long Strait by the shallow bathymetry gradients, westward winds, and the SSH gradient centered at one end on the ESS.  Note that the coastal current that flows southeastward in the southwestern Chukchi Sea [the Siberian Coastal Current (SCC)] is seasonal and it has been observed only during some years (Weingartner et al. 1999).  The model does show a similar southeastward flow (SCC) sometimes and at shorter time scales. However, the long-term mean flow is to the northwest through Long Strait. This flow in the mean model results follows the topographic constraints of the Siberian coast and is a result of the predominate westward winds and pressure head forcing, in agreement with the cited literature. It's seasonal and interannual variability are driven by the combination of these two factors.

Coachman, L. K., & Aagaard, K. (1966). On the water exchange through Bering Strait. Limnology and Oceanography, 11(1), 44–59. https://doi.org/10.4319/lo.1966.11.1.0044

Peralta-Ferriz, C., & Woodgate, R. A. (2017). The dominant role of the East Siberian Sea in driving the oceanic flow through the Bering Strait—Conclusions from GRACE ocean mass satellite data and in situ mooring observations between 2002 and 2016. Geophysical Research Letters, 44, 11,472– 11,481. https://doi.org/10.1002/2017GL075179

Stigebrandt, A. (1984). The North Pacific: A global-scale estuary. Journal of Physical Oceanography, 14(2), 464–470. https://doi.org/10.1175/1520-0485(1984)014%3C0464:TNPAGS%3E2.0.CO;2

Weingartner, T. J., Danielson, S., Sasaki, Y., Pavlov, V., Kulakov, M.: The Siberian Coastal Current. A wind- and buoyancy-forced Arctic coastal current, J. Geophys. Res.-Oceans, 104, 29697– 29713, 1999.

Woodgate, R.A., Aagaard, K., Weingartner, T.J., 2005. A year in the physical oceanography of the Chukchi Sea: moored measurements from autumn 1990–1991. Deep Sea Res. II 52, 3116– 3149.